# Genome-Wide Identification and Expression Analysis of *Rosa roxburghii* Autophagy-Related Genes in Response to Top-Rot Disease

**DOI:** 10.3390/biom13030556

**Published:** 2023-03-17

**Authors:** Kaisha Luo, Jiaohong Li, Min Lu, Huaming An, Xiaomao Wu

**Affiliations:** 1Guizhou Engineering Research Center for Fruit Crops, College of Agriculture, Guizhou University, Guiyang 550025, China; 2Institute of Crop Protection, College of Agriculture, Guizhou University, Guiyang 550025, China; 3The Provincial Key Laboratory for Agricultural Pest Management of Mountainous Region, Guiyang 550025, China

**Keywords:** autophagy, *Rosa roxburghii*, *Colletotrichum fructicola*, calcium, top rot

## Abstract

Autophagy is a highly conserved process in eukaryotes that degrades and recycles damaged cells in plants and is involved in plant growth, development, senescence, and resistance to external stress. Top-rot disease (TRD) in *Rosa roxburghii* fruits caused by *Colletotrichum fructicola* often leads to huge yield losses. However, little information is available about the autophagy underlying the defense response to TRD. Here, we identified a total of 40 *R. roxburghii* autophagy-related genes (*RrATGs*), which were highly homologous to *Arabidopsis thaliana ATGs*. Transcriptomic data show that *RrATGs* were involved in the development and ripening processes of *R. roxburghii* fruits. Gene expression patterns in fruits with different degrees of TRD occurrence suggest that several members of the RrATGs family responded to TRD, of which *RrATG18e* was significantly up-regulated at the initial infection stage of *C. fructicola*. Furthermore, exogenous calcium (Ca^2+^) significantly promoted the mRNA accumulation of *RrATG18e* and fruit resistance to TRD, suggesting that this gene might be involved in the calcium-mediated TRD defense response. This study provided a better understanding of *R. roxburghii* autophagy-related genes and their potential roles in disease resistance.

## 1. Introduction

*R. roxburghii* is a medicine and food homologous crop, whose fruits are rich in vitro antioxidant substances beneficial to human health, such as total phenols, flavonoids, triterpenes, and L-ascorbic acid [1]. Accordingly, *R. roxburghii* has been widely cultivated as an economic crop in Southwest China, especially in Guizhou Province, where the cultivated areas have so far exceeded 140,000 hm^2^ [1]. In recent years, there has been a new fungal disease named TRD in *R. roxburghii* fruits caused by *C. fructicola* [2]. At the occurrence beginning of this disease, there were obviously small, dark red diseased spots at the junction of the top fruit pulp and sepals, whereas in the later developmental stage, the pulp became dark brown and rotten, and the fruit was highly prone to drop at pre-harvest. TRD has caused serious yield losses and quality declines in *R. roxburghii* production in China every year [2].

Plants growing in nature are exposed to many adverse biotic and abiotic stresses such as drought, cold, salt, and pathogens. Unfortunately, they cannot choose their desired survival environments by moving, hence they have evolved a sophisticated immune system to fight against various stresses [3,4]. Based on how the immune response is triggered, the innate immune system of plants is divided into two categories: pattern-triggered immunity (PTI) and effector-triggered immunity (ETI) [5,6]. ETI is a plant-specific defense response that can accelerate and amplify the PTI response and trigger the hypersensitive response (HR) to cell death of the host cell to stop the pathogen from multiplying [3,7]. This cell death linked to genetics might be essential for resistance to plant diseases [8]. ‘Autophagic cell death’ is often defined as a type of cell death by morphological criteria, which would contribute to inhibiting mycelia elongation, especially when the invader is a biotrophic pathogen [7,8].

The main structure of autophagy is the autophagosome, which can form a double-membrane structure to engulf damaged or unwanted macromolecules/organelles to deliver into the vacuole or lysosome for degradation or recycling [9,10]. From yeast to animals and plants, the autophagic process is highly conserved in eukaryotes and controlled by autophagy-related genes (*ATGs*) [11,12,13]. Based on the roles that core ATG proteins play in the autophagic pathways, ATG proteins are approximately divided into the following functional groups: (i) the ATG1/13 kinase complex; (ii) the PI3K kinase complex; (iii) the ATG9 kinase complex; and (iv) ATG8-PE and ATG12-5 ubiquitination-like conjugation systems [10,14,15]. Up to now, there have been more than 35 *ATGs* identified in yeast and *Arabidopsis*, and many homologous *ATGs* have been identified in other species based on the model plants [10,16,17,18,19,20,21,22,23,24,25,26]. Subsequently, the functions of these *ATGs* in the plant were gradually demonstrated. For example, *OsATG8a/8b* improved nitrogen uptake and utilization, contributing to improving the rice grain quality and yield [27,28]. *MdATG5a* and *MdATG10* could enhance the drought/salt stress tolerance of apple plants, and *MdATG3b* exhibited better growth performance as the nutrient supply was limited [29,30,31]. Besides, *ATGs* have also been demonstrated to play an important role in the resistance of plants to pathogens. For instance, the silencing of *PbrATG8c* decreased the resistance to *Botryosphaeria dothidea* in pear leaves [32]. In *Arabidopsis*, phosphorylation of *AtATG18a* compromised the resistance of plants to *Botrytis cinerea*, whereas overexpression of the *AtATG18a* dephosphorylation-mimic form promoted the accumulation of autophagosomes and increased plant resistance to *B. cinerea*, as well as overexpressing *AtATG5*, *AtATG7*, and *AtATG8a* enhanced plant resistance to necrotrophic pathogens [33,34,35]. Moreover, *MaATG8s* were essential for the resistance of banana plants to Fusarium wilt [22]. In brief, autophagy plays a key role in host–pathogen interactions. However, there is little information available about autophagy-related genes of *R. roxburghii* in response to TRD.

Ca^2+^ signaling is one of the important transduction events in plant immunity. Sufficient external Ca^2+^ is indispensable for transmitting the perception of nonself signals to an intracellular signaling pathway and is also essential for activating antimicrobial responses to inhibit the growth of pathogens [36]. When *Arabidopsis* plants were grown in a low Ca^2+^ medium, the reduction of PTI responses was examined [37]. Similarly, the high-vigor maize seeds grew better without being affected by pathogens due to the higher concentration of free Ca^2+^ in the cytoplasm and nucleus [38]. Recently, the most prevalent approach to reducing the damage caused by various kinds of stresses was to increase intracellular Ca^2+^ concentration through the application of exogenous calcium salt [39]. Spraying calcium chloride before pathogenic inoculation could enhance the resistance of pear leaves to *B. dothidea* [40]. The application of calcium chloride was advantageous in controlling *Phytophthora pistaciae* gummosis in commercial pistachio crops [41]. Foliar spraying of exogenous calcium could reduce ozone damage in rice and had better control effects against apple fruit watercore [42,43].

TRD caused by *C. fructicola* has been one of the most serious diseases of *R. roxburghii* [2]. *Colletotrichum* spp. is one of the top 10 fungal pathogens from the international community and causes enormous yield losses to crops every year [44]. At present, chemical fungicides are an effective way to control TRD. However, the long-term use of chemical fungicides inevitably results in potential adverse health effects on ecological environments, wildlife populations, and humans due to their hazardous nature of toxic residues. As a consequence, an environmentally friendly alternative to chemical fungicides needs to be taken into consideration for safely controlling these diseases and potential issues of concern. In this study, a total of 40 *RrATGs* were identified and were further verified to take part in the response of *R. roxburghii* fruits to TRD, of which *RrATG18e* was significantly up-regulated at the initial infection stage of *C. fructicola* under the condition of exogenous Ca^2+^. This study would provide new insight into the potential role of *RrATGs* in the defensive responses of *R. roxburghii* fruits to TRD.

## 2. Materials and Methods

### 2.1. Genome-Wide Identification of ATG Family Genes in R. roxburghii

The corresponding protein sequence of AtATGs was downloaded from the TAIR database (https://www.arabidopsis.org/index.jsp (accessed on 3 March 2022)) based on the ID number of known AtATGs, and a BLASTP was performed with the existing genome of ‘Guinong 5’ on TBtools (v.1.098769) software, with the E-value set to 1 × 10^−20^, NumofHits set to 5, and NumofAligns was also set to 5 to filter the results. Subsequently, the conserved domain of the candidate protein sequence was analyzed and identified by Pfam at the website of http://pfam.xfam.org/ (accessed on 4 March 2022). All the identified genes were named *RrATGs*. The amino acid length, molecular weight, and theoretical isoelectric point of proteins of RrATGs were obtained using BioXM2.6. The subcellular localization was predicted using the online tool WoLF PSORT (https://wolfpsort.hgc.jp/ (accessed on 23 September 2022)).

### 2.2. Bioinformatics Analysis of RrATGs

Using the ATG protein sequences of *Arabidopsis thaliana*, *Nicotiana tabacum*, *Oryza sativa*, *Vitis vinifera*, and the putative RrATG proteins, a total of five species sequences were submitted to ClustalW for the multiple sequence alignment. The generated file was used to construct a phylogenetic tree through the neighbor-joining method, and bootstrap analyses were carried out in MEGA 7 software (in 1000 replicates). The bootstrap value below 50% was not displayed in the phylogenetic tree. The chromosome localization and collinear analysis were conducted by TBtools (v.1.098769) software. The exon–intron structure of *RrATGs* was visualized using the GSDS v2.0 (http://gsds.gao-lab.org/ (accessed on 25 September 2022)) online website and put together based on the different functional groups.

### 2.3. Plant Materials and Treatments

The fruits with different degrees of TRD occurrence were taken from *R. roxburghii* plants with tree years of 10 in the orchards of Chaxiang Village, Gujiao Town, Longli County, Guizhou Province, China, in 2022 (26°54′ N, 106°95′ E). The healthy and diseased fruits were ranked according to the proportion of fruit spot size to the total surface area of the fruit as follows [45]: grade 0 is no incidence, grade 1 is 1–10%, grade 2 is 11–25%, grade 3 is 26–50%, and grade 4 is >50%. The sampling site was the junction of 1 cm of diseased spot and healthy flesh fruit tissue, i.e., 0.5 cm of diseased flesh fruit tissue and 0.5 cm of healthy flesh fruit tissue. Samples were frozen at −180 °C in an ultra-low temperature refrigerator for RNA extraction.

The pathogen inoculation trials were conducted in the fruit germplasm repository of Guizhou University, Guizhou, China, in 2022 (26°42.408′ N, 106°67.353′ E). Twelve-year-old plants of ‘Guinong 5’ *R. roxburghii* were selected as in vivo materials. Considering the TRD occurrence period of *R. roxburghii* fruits in the field, healthy fruits with uniform size were selected to inoculate *C. fructicola* on July 13. Firstly, the fruit surface was disinfected with 75% ethanol for 15 min and then washed with sterile water. Subsequently, the fruits were sprayed with 2% calcium acetate (Ca^2+^). Controls were sprayed with an equal amount of double-distilled water (H_2_O). After 24 h of spraying, the fruits were in vivo wound-inoculated near the sepal end with strain CXCDF-3 activated on potato dextrose agar (PDA) using a pre-prepared sterile needle. Controls were in vivo wound-inoculated with sterilized PDA. A total of 400 fruits were treated with Ca^2+^ or H_2_O (control). Ten fruits were sampled for each plot (in three replicates) at thirteen days after inoculation (DPI). The spot area of the diseased fruit was calculated using the elliptical area formula. The tissue (1 cm) was taken from the edge of the spot in the diseased fruit. The tissue samples were immediately transported back to the laboratory and frozen in liquid nitrogen at −180 °C.

### 2.4. RNA-Seq Analysis of RrATGs Tissue-Specific Expression

Based on the databases of genomic RNA-seq [46,47], the tissue-specific expression profiles of all the identified *RrATGs* in four different tissues (stem, leaf, flower, and fruit) and in *R. roxburghii* fruit at different developmental stages were analyzed. The heat map was also plotted using TBtools (v.1.098769).

### 2.5. RNA Extraction, cDNA Synthesis, and qRT-PCR Analysis

Total RNA was extracted with the RNAprep Pure Plant Kit (Tiangen Biotech Co., Ltd., Beijing, China). RNA integrity was evaluated using agarose gel electrophoresis and a NanoDrop spectrophotometer (Thermo Scientific, Los Angeles, CA, USA). A total of 1 μg high-quality RNA was used as the input material for cDNA synthesis with the PrimeScrip RT Reagent Kit with gDNA Eraser (Perfect Real Time) (TaKaRa, Inc., Dalian, China). Real-time quantitative PCR (qRT-PCR) was implemented on the ABI ViiA 7 DX system (Applied Biosystems) using TB Green Premix Ex Taq II (TaKaRa) with the ubiquitin (*UBQ*) gene as a reference gene to normalize expression data. The specific primer used for qRT-PCR was designed using Primer Premier 5 software and the sequence is listed in Appendix A. Each PCR reaction contained 5.0 µL TB Green mix, 0.8 µL primers, and 1.0 µL diluted cDNA in a final volume of 10 µL. The amplification conditions were as follows: 30 s of denaturation at 95 °C, followed by 40 cycles at 95 °C for 5 s and 60 °C for 30 s, then 95 °C for 15 s, 60 °C for 1 min, 95 °C for 15 s. Each experiment was repeated at least triplicate and each gene was calculated with the 2^−ΔΔCT^ method for the relative expression.

### 2.6. Total Calcium Content Detection

Ca^2+^ content (mmol·L^−1^) was measured using a calcium colorimetric assay kit (Beyotime Biotechnology Co., Ltd., Shanghai, China) [48].

### 2.7. Statistical Analysis

Experimental data were expressed as the mean ± standard deviation (*SD*) of three independent replicates. Data were analyzed by analysis of variance (ANOVA), and means were compared using Tukey’s multipole difference test (*p* < 0.05). All statistical analyses were implemented with the SPSS 20.0 statistical package (IBM SPSS Statistics). Gene expression heat maps were drawn using TBtools software (v.1.098769).

## 3. Results

### 3.1. Identification of 40 ATGs in R. roxburghii

Based on the known AtATG amino acid protein sequences as queries, a total of 40 putative RrATGs were identified from the genome of *R. roxburghii*. These RrATGs showed 49.16% to 91.45% of their sequence identified with AtATGs and had close phylogenetic relationships with other species having homologous ATGs (Table 1 and Figure 1). The RrATG family identified a total of 19 subfamilies. In the RrATG subfamilies, the RrATG2/3/4/5/6/7/9/10/16/20/101, and the RrVPS15/34 had only one member, whereas other subfamilies contained multiple members: seven members in the RrATG18 subfamily, five members in the RrATG8 subfamily, four members in the RrATG1 subfamily and the RrATG12 subfamily, respectively, three members in the RrATG5 subfamily and RrATG13 subfamily, respectively, and two members in the RrTOR subfamily. In contrast to AtATGs, RrATGs were identified with more genes in RrATG5, RrATG12, RrATG13, and RrTOR, but less in the RrATG4, RrATG8, and RrATG18 subfamilies. In addition, bioinformatics analysis results indicate that the length of amino acids ranged from 77 to 2459 aa and the molecular weights ranged from 8.73 to 276.22 kD. The RrTORa possesses the maximum amino acids and molecular weights of all RrATG proteins. This information suggests the RrATGs identified might exist in significant variations with potential functional differentiation. The prediction of subcellular location results shows that most RrATGs were predicted to localize to the cytoplasm and nucleus, accounting for more than 50%, followed by mitochondria, chloroplasts, Golgi, plasma membrane, endoplasmic reticulum, extracellular, and the cytoskeleton. In addition to the RrATG1 subfamily, there are differences in the subcellular localization of the RrATG subfamilies, which contain several members, especially the RrATG18 subfamily, which has seven genes that are not in the same location. The significantly various subcellular localization could mean that they played various roles in the autophagic process.

### 3.2. Bioinformatics Analysis of RrATGs

To assess the evolutionary relationships of RrATGs we used *R. roxburghii*, *Arabidopsis thaliana*, *Nicotiana tabacum*, *Oryza sativa*, and *Vitis vinifera* autophagy-proteins to construct the neighbor-joining phylogenetic trees. As shown in Figure 1, most RrATGs were clustered in one branch and showed close homology with *Vitis vinifera*. Some multiple members of the RrATG subfamily were clustered in one branch (RrATG1, RrATG5, and RrATG12 superfamily), containing the least identity value RrATG1d. Whereas other multiple members were clustered in two or three branches. For instance, the RrATG13 subfamily with five members was clustered in two branches, and the RrATG18 subfamily with seven members was clustered in three branches. In total, RrATGs had similar evolutionary relationships but were not consistent with function. Furthermore, chromosome location showed that seven chromosomes distributed all *RrATGs*, whose size was indicated by their relative length (Figure 2). Chromosome 6 (chr6) contained the greatest number of *RrATGs* (9); the minimum was chromosome 3 (chr3), which only contained two genes. In addition, the multiple members of the *RrATG* subfamilies were not localized on the same chromosome except for the *RrTOR* subfamily. The *RrATG12* subfamily has four members spread over four chromosomes. Segmental duplication events play an important role in the evolution of the family. As shown in Figure 2, five pairs of genes were predicted to be segmental duplications, accounting for about 25% of all *RrATGs*. The *RrATG8b* and *RrATG8c*, *RrATG8b* and *RrATG8d*, *RrATG8c* and *RrATG8d*, *RrATG13b* and *RrATG13c*, and *RrATG18g* and *RrATG18h* had collinear correlations, which were linked with red color, respectively. The chromosomal distribution and segmental duplication provided further evidence for the wide functional divergence.

The exon–intron structure of *RrATGs* was predicted by the GSDS v2.0 online website, as shown in Figure 3a. Fifty-five exons were found in *RrTORa*, fifty-four in *RrTORb*, and other genes ranged from one to nineteen. Moreover, all *RrATGs* were predicted to be introns except for *RrATG12b*. Given the importance of conserved domains for assessing protein function, the SMART program was used to visualize the conserved domains of RrATG proteins (Figure 3b). From the predicted visualization, we discovered that a lot of RrATGs had their own ATG domains, for example, RrATG5, RrATG6, RrATG7, RrATG8, RrATG9, RrATG12, RrATG13, and RrATG101. Besides, the same functional group in the RrATGs subfamily usually contains similar conserved domains. Serine/threonine protein kinases emerged from the RrATG1 subfamily, and WD40 domains existed in all members of the RrATG18 subfamily. However, the members of the RrATG18 subfamily were still divided into two groups because the special breast carcinoma amplified sequence 3 (BCAS3) was only encoded by RrATG18g and RrATG18h. Notably, RrATG18e has a specific DIOX_N conserved domain, which means it may have a particular protein function. Additionally, the phox homology (PX) domain, PI3K, chorein N, peptidase C54, peptidase C78, ThiF, RHOD, Hydrolase 4, and DUF3385 were depicted in RrATGs. The number of conserved domains indicates that each RrATG protein may play various roles in regulating autophagy processes.

### 3.3. RNA-Seq Analyses of RrATGs in Different Tissues and Developmental Stage-Specific Expressions

To understand the importance of *RrATGs* in a plant’s growth and development, the expression levels of 40 *RrATG*s in various tissues (flower, leaf, stem, fruit), and at different fruit developmental stages (30, 60, 90, and 120 days after anthesis) were retrieved from the genomic RNA-seq databases. As exhibited in Figure 4, in which *RrATG1b*, *RrATG5b*, *RrATG8a*, *RrATG12a*, *RrATG12d*, and *RrATG18b* displayed extremely low relative expression levels in every tissue mentioned, while other genes in the same *RrATG* subfamily showed higher expression levels. This suggests that the members of the same subfamily had significant tissue specificity, implying that they had functional differences. Among the four different tissues, no *RrATGs* had the highest expression in flowers, suggesting that *RrATGs* were less involved in the developmental process of flowers. *RrATG1b* and *RrATG10* had higher expression in leaves than in other tissues; probably they played more roles in leaf development than other tissues. *RrATG4*, *RrATG5b,* and *RrATG5c* had higher expression in the stem, while the remaining genes were expressed higher in fruits, indicating that most *RrATGs* were more involved in the ripening process of fruits. Different *RrATGs* were differentially expressed at different developmental stages of fruits. *RrATG1a*, *RrATG101*, and *RrTOR* were more highly expressed in fruits 30 days after anthesis, indicating that they were mainly involved in the development of young fruits. As well as the *RrATG3*, *RrATG7*, *RrATG16*, *RrATG20*, and *RrVPS34*, most members of the *RrATG8/18* subfamily were expressed centrally during mid-fruit development. The *RrATGs* mainly involved in fruit ripening were members of the *RrATG12* and *RrATG13* subfamilies. In conclusion, these data suggest that *RrATG*s had tissue-specific and spatiotemporal expression properties, which were involved in the growth and developmental processes of *R. roxburghii*, mediating the ripening process of fruits.

### 3.4. Expression Profiles of RrATGs with Different TRD Grades of R. roxburghii Fruits

To explore the mechanism of *RrATGs* in response to the pathogenesis of *R. roxburghii* TRD, the expression levels of *RrATGs* in fruits with different grades of TRD were evaluated. The q-PCR results are presented in Figure 5b. Using the expression in healthy fruits (0 grade) as a template, the expression of most *RrATG*s was significantly decreased after infection by *C. fructicola,* except for *RrATG18e*, which was significantly up-regulated at the early stage of fruit susceptibility. This indicates that *RrATG18e* might play a key role in the early resistance to TRD. The expression of *RrATG18e* decreased slowly but was still significantly higher than other genes as the fruit disease progressed, suggesting that *RrATG18e* plays a central role in response to *C. fructicola* infection in *R. roxburghii* fruits. In addition, the relative expression of some *RrATGs* also deserves our attention. The expression of *RrATG5c*, *RrATG18d,* and *RrATG18h* in different grades of TRD fruits showed similar trends, with a slight decrease followed by an increase, of which *RrATG18d* showed a greater increase in expression in grade 4 TRD fruits. The expression of the *RrATG4* gene did not decrease significantly in grade 1 TRD fruits, and its expression only decreased slowly with the increase in disease index. The expression of *RrATG12b*, *RrATG12c*, *RrATG9*, *RrATG13b,* and *RrATG13c* rose in grade 3 TRD fruits compared to grade 2 TRD fruits. While RrATG1 subfamily members and *RrATG10* genes were particularly low in expression in grades 1–4 TRD fruits, other genes with floating decreasing expression would be minimally expressed in grade 4 TRD fruits. The above trends in gene expression indicate that different *RrATG* genes responded differently in different grades of TRD fruits.

### 3.5. Field Control Effect of 2% Calcium Acetate (Ca^2+^) against TRD in R. roxburghii Fruits

The previous study had shown that the application of exogenous Ca^2+^ could inhibit the infection of *B. dothidea* in pear leaves [40]. The same results were obtained when exogenous Ca^2+^ was applied to enhance the resistance of *R. roxburghii* fruits to *C. fructicola* infection. As shown in Figure 6, the area of disease spots of Ca^2+^-treated fruits was significantly lower than that of H_2_O-treated fruits after the fruits were infected with *C. fructicola* at 13 DPI (Figure 6c). At the same time, the total calcium content levels showed that the fruits with 2% Ca^2+^ treatment were significantly higher than those of the H_2_O-treated fruits (Figure 6d).

### 3.6. Expression Profiles of RrATGs under C. fructicola Infection after Ca^2+^ Treatment

To further explore the potential relationship between Ca^2+^, autophagy, and *C. fructicola* infection in *R. roxburghii* fruits, the changes in the expression patterns of *RrATGs* after Ca^2+^ and H_2_O treatment at 13 DPI were investigated (Figure 6b). The fruits inoculated with control PDA after treatment with H_2_O were used as a template. The expression levels of several *RrATGs* were highly induced when Ca^2+^-treated fruits were inoculated with control PDA, such as *RrATG1/2/3/7/18/TOR/VPS34*, suggesting that exogenous Ca^2+^ could stimulate the expression of some *RrATGs* even though fruits were not infected by *C. fructicola*. When the fruits were inoculated with *C. fructicola*, *RrATG5c* and *RrATG18e* gene expression was significantly up-regulated, and *RrATG8b*/12b/12c/13a/13b/13c/18b/8d/8f*/20* were significantly down-regulated under H_2_O treatment. And Ca^2+^-treatment promoted not only a significant increase in the expression of *RrATG5c* and *RrATG18e* but also promoted the expression of *RrATG4/6/7/9/10* genes. However, there was no significant effect on the down-regulated expression of genes when fruits were inoculated with *C. fructicola* under both Ca^2+^- and H_2_O-treated conditions. Notably, *RrATG18e* expression was sharply up-regulated by 11-fold in fruits inoculated with *C. fructicola* under Ca^2+^ treatment. Therefore, *RrATG18e* might play a comparatively important role in calcium-mediated enhancement of the resistance of *R. roxburghii* fruits to TRD.

## 4. Discussion

ETI is the innate immune system of plants, which can enhance plant resistance by effector recognition of pathogenic motifs attached to the cell surface, and leads to the hypersensitive response (HR) to control mycelia growth [49]. Autophagic cell death may be the result of an overactive defense response of HR during the development of resistance, meaning that cell death is essential for resistance, and it makes better sense especially when the invader is *C. fructicola* (a biotrophic pathogen that prefers a living host) [8]. Accordingly, there was an attempt to understand how autophagy responds when *R. roxburghii* fruits are infected by *C. fructicola*. First of all, there were 41 AtATGs used as queries to identify 40 RrATGs, except that RrATG11 was not identified. Among many different species, the ATG1, ATG8, and ATG18 subfamilies were identified as having multiple members: AtATG1/8/18 had 4/9/8 members; CsARG1/8/18 had 2/5/6 members; MtATG1/8/18 had 3/8/8 members; CsATG1/8/18 had 4/8/8 members; VvATG1/8/18 had 2/6/7 members; NtATG1/8/18 had 3/5/6 members; OsATG1/8/18 had 3/7/6 members; and ZmATG1/8/18 had 4/5/9 members [14,17,18,19,23,24,25,26]. Similarly, 4, 5, and 7 members were respectively identified in RrATG1, 8, and 18 subfamilies. Besides, the conserved domains of members of the same subfamily are also extremely similar in these different species. ATG1s encode serine/threonine protein kinases family. ATG8 domain is ubiquitin homologs (UBQ) in the ATG8 subfamily. ATG16 consists of multiple WD40 domains. ATG20 contains a phox homology (PX) domain. VPS15 possesses both the S_TKc domain and the WD40 domain, the former being the structural domain of the protein kinase family that catalyzes protein phosphorylation. VPS34 is the phosphatidylinositol 3 kinase (PI3K) family. TOR, as a conserved phosphatidylinositol kinase-associated protein kinase, always contains a specific rapamycin-binding domain (DUF3385). The important RrATG18 subfamily contains the WD40 structural domain in all members. Like other species, it can be divided into two categories according to the presence or absence of the BCAS3 domain at the C-terminal. These similarities obviously indicate the ATG family remains highly conserved over a long evolutionary period. In the collinearity analysis, *RrATGs* were found to have a total of 5 collinearity gene pairs. The members of the *RrATG8* and *RrATG18* subfamilies had a large number of segmental duplication events, indicating those members belonging to their subfamily were mostly derived from gene duplication during evolution [24,26,50].

The importance of autophagy has been widely reported in nutrient cycling. Leaf-senescence-induced autophagy occurs to fully recirculate 75% of the nitrogen stored in the chloroplast [15]. *OsATG8a/8b* improved rice grain yield and quality by enhancing nitrogen uptake and utilization; *AtATG18a* was induced and expressed under sucrose and nitrogen starvation during the senescence of *Arabidopsis thaliana* leaves [27,28,51]. Our transcriptome data analysis shows that the most *RrATGs* were highly expressed in the middle and late stages of fruit development, which might be the involvement of *RrATGs* in the development and ripening processes of *R. roxburghii* fruits through nutrient allocation or material recirculation. Normally, when *Colletotrichum* appressoria penetrate fruits, their mycelia first attach to the cuticle and uppermost epidermal cell layers of immature fruits to develop and fully erupt when the fruits ripen [52]. The differential expression of the *RrATGs* with the increased TRD occurrence of fruits under the conditions of nature suggests that different response mechanisms may occur in *RrATG* genes faced with stress. Previous studies had reported that the overexpression of *MdATG5a* and *MdATG10* enhanced the drought/salt stress tolerance of apple plants, and over-expressed *MdATG3b* displayed better growth performance when nutrient supplies were limited [29,30,31]. The silencing of *PbrATG8c* decreased the resistance to *B. dothidea* in the pear [32]. Likewise, *RrATG18e* responded positively to *C. fructicola* infection in different degrees of TRD occurrence, especially at the early infection stage, suggesting that *RrATG18e* might be a potential key gene for improving the resistance of *R. roxburghii* fruits to TRD.

Ca^2+^ signaling events are important transduction events in plant immunity [36]. When a plant initially receives the signal of pathogen infection, the higher intracellular Ca^2+^ concentration can induce an immediate and strong defense response to enhance the plant’s resistance. The most prevalent approach is to apply exogenous calcium salt to plant trees [39]. Since TRD of *R. roxburghii* caused by *C. fructicola* usually occurs sporadically in early July and in vitro culture tests of *C. fructicola* have shown that it is suitable for the growth of TRD at 25 °C [2], the field trials were thus conducted in July. Higher ambient temperatures during the trial may be more favorable for pathogen infection. In this study, the area of disease spots was smaller and the total calcium contents were higher in Ca^2+^-treated fruits compared with H_2_O-treated fruits, which might be due to the spraying of exogenous calcium increasing the intracellular Ca^2+^ concentration to enhance the innate immune response and inhibit mycelia growth. In addition, the expression of *RrATG4/5c/6/7/9/10/18e* was significantly higher after Ca^2+^ treatment than H_2_O treatment under *C. fructicola* infection, indicating these genes might be the core *RrATGs* in response to the Ca^2+^-mediated TRD defense mechanism. Finally, the high expression of *RrATG18e* in both naturally diseased and *C. fructicola*-inoculated fruits raised our concern. A positive response of *RrATG18e* to early *C. fructicola* infection was clearly observed in natural fruits of TRD, and the mRNA of *RrATG18e* was highly accumulated in fruits inoculated with *C. fructicola* at 13 DPI after Ca^2+^ and H_2_O treatments, suggesting that *RrATG18e* may be a core autophagy gene in the autophagic pathway to enhance *R. roxburghii* resistance to TRD. Moreover, the expression of *RrATG18e* was also induced higher after Ca^2+^ treatment under inoculating the control PDA, indicating that *RrATG18e* could be stimulated by exogenous Ca^2+^. In conclusion, *RrATG18e* would be the core *RrATGs* involved in the calcium-mediated TRD defense response.

AtATG18a protein is critical for autophagosome formation, and its phosphorylation and overexpression have also been shown to play a key role in plant resistance to pathogenic infection [34,51,53]. From the phylogenetic tree, we can see RrATG18e and AtATG18a were constructed on a branch, which suggests they have a close evolutionary relationship and maybe have a similar function. Besides, unlike other members of the RrATG18 subfamily, RrATG18e was predicted in the cytoplasm and had a specific DIOX_N conserved structural domain. Under normal conditions, the free Ca^2+^ stored in the extracellular space and certain intracellular stores is 10,000-fold higher than the resting cytoplasmic free Ca^2+^ level. Such a Ca^2+^ concentration gradient allows pathogens to infect plants by triggering intracytoplasmic Ca^2+^ spikes. Next, it will transmit immune signals to downstream cellular responses through a decoding mechanism formed by Ca^2+^ sensors [36]. In addition, the germination of highly viable maize seeds was not affected by *F. graminearum* infection because it probably had a higher concentration of free Ca^2+^ in the cytoplasm of the embryonic cells [38]. Moreover, the dependent function of AtATG18a in the cytoplasm was sufficient to induce autophagy and enhance resistance against *B. cinerea* [34]. Therefore, the above observations suggest that the excellent performance of *RrATG18e* against *C. fructicola* under Ca^2+^ treatment may be related to its being localized in the cytoplasm. However, relevant validation remains to be done in subsequent experiments. Further, the special DIOX_N unique to the RrATG18e may be the key function domain for the defense TRD. This is a highly conserved N-terminal region of proteins with 2-oxoglutarate/Fe(II)-dependent dioxygenase activity and is widely distributed in nature [54]. It can promote the accumulation of flavonoids and positively regulate plant abiotic stress tolerance [55]. Perhaps the response of *RrATG18e* to *C. fructicola* infection in *R. roxburghii* fruits is closely related to this special conserved domain.

## 5. Conclusions

A total of 40 *RrATGs* were identified in *R. roxburghii*, and bioinformatic analysis shows that they were highly homologous to *Arabidopsis thaliana* autophagy-related genes (*AtATG*s). Most *RrATGs* were expressed up-regulated in *R. roxburghii* fruits at the medium to late stages of fruit development and down-regulated in fruits with TRD. Exogenous Ca^2+^ treatment enhanced the *R. roxburghii* fruit resistance to TRD and promoted the mRNA accumulation of *RrATGs*, of which the highest expression levels of *RrATG18e* suggest that it might be the core *RrATGs* involved in the calcium-mediated TRD defense response. In this study, *RrATGs* were analyzed and initially revealed their involvement in response to *C. fructicola* infection, which laid the foundation for further studies on the molecular mechanism of *R. roxburghii* resistance to TRD. Further studies are needed to understand the physiological functions of *RrATG18e* in *R. roxburghii’s* resistance to TRD.

## Figures and Tables

**Figure 1 biomolecules-13-00556-f001:**
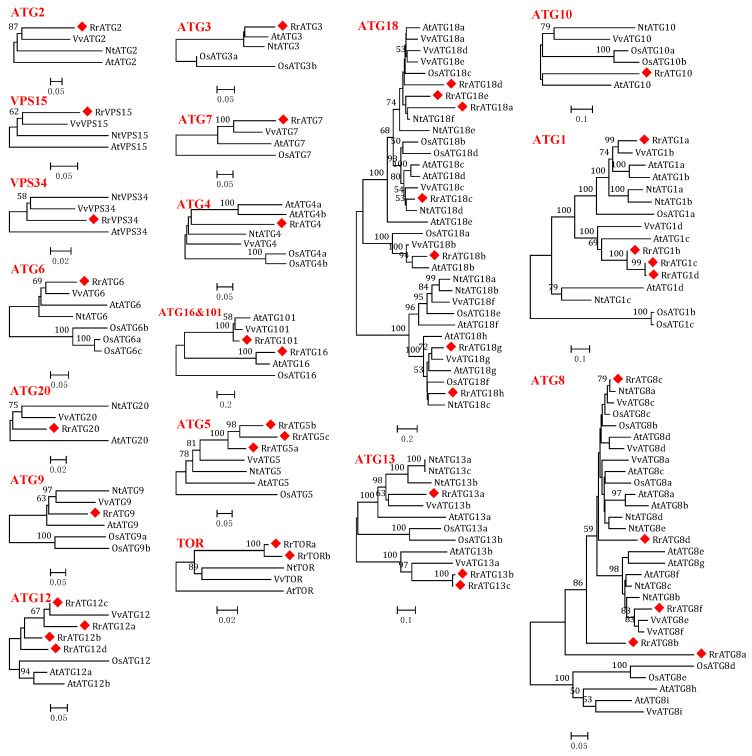
Phylogenetic trees were constructed using the neighbor-joining method with 1000 bootstrap values for ATG protein sequences of five species: *R. roxburghii*, *Arabidopsis thaliana*, *Vitis vinifera*, *Nicotiana tabacum*, and *Oryza sativa*. *RrATG*s are marked with red squares and bootstrap values below 50% are not shown in the phylogenetic trees.

**Figure 2 biomolecules-13-00556-f002:**
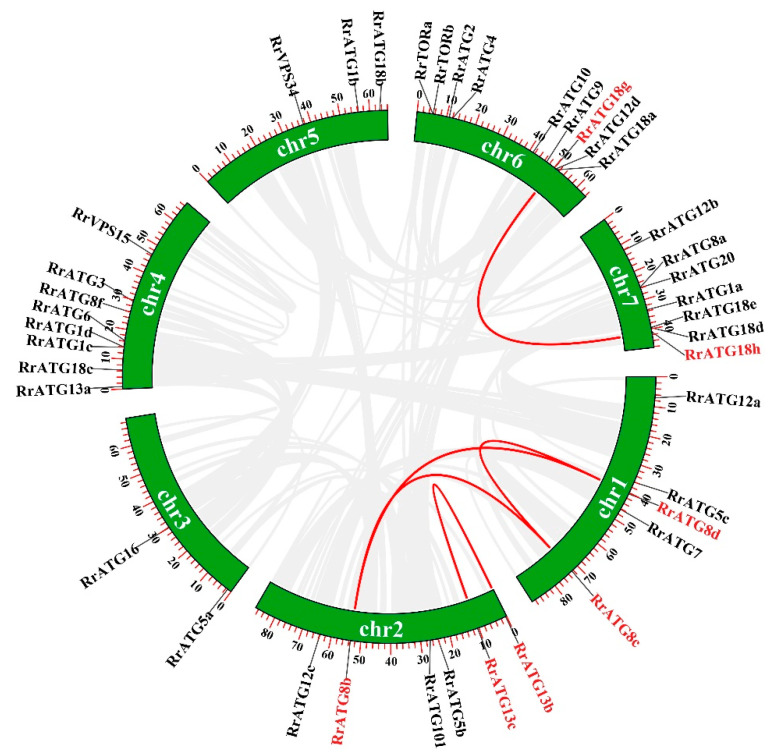
Chromosome distribution and gene replication of *RrATG* genes. Genes that had segmental duplications were highlighted in red and pairwise collinearity was linked by red lines.

**Figure 3 biomolecules-13-00556-f003:**
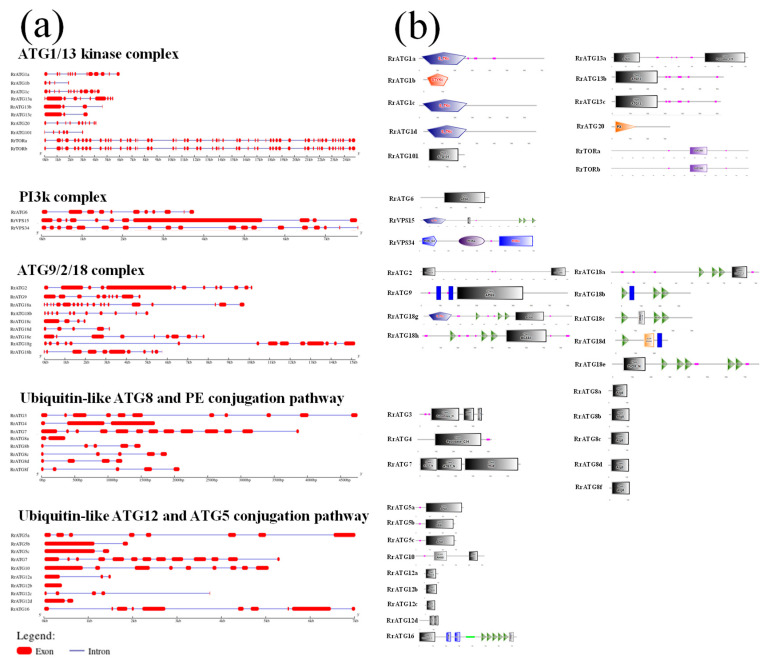
Gene structures and conserved domains of RrATGs. (**a**) Exon–intron of *RrATGs*, exons indicated by red squares and introns indicated by blue lines. (**b**) The predicted proteins conserved domain of RrATGs.

**Figure 4 biomolecules-13-00556-f004:**
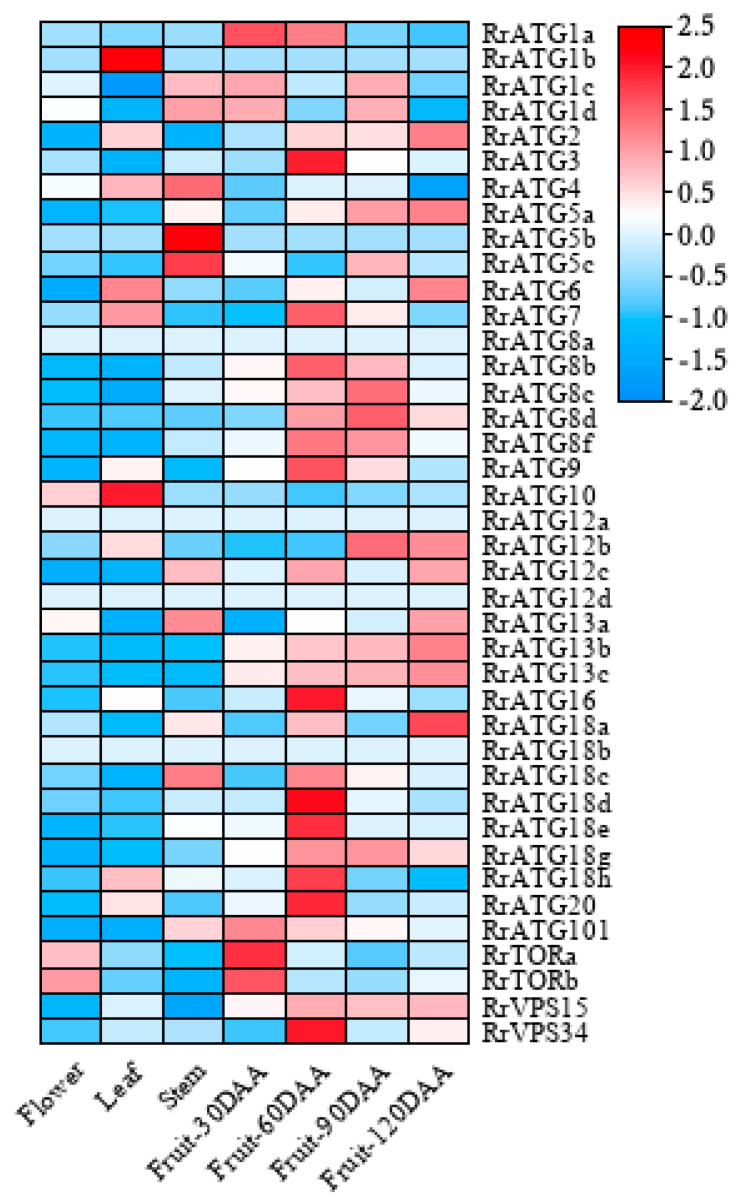
Expression patterns of *RrATGs* in different tissues and different fruit developmental stages. Raw data from genomic RNA-seq databases in *R. roxburghii*. DAA was represented days after anthesis. Red and blue represent the higher and lower expression levels in each row, respectively.

**Figure 5 biomolecules-13-00556-f005:**
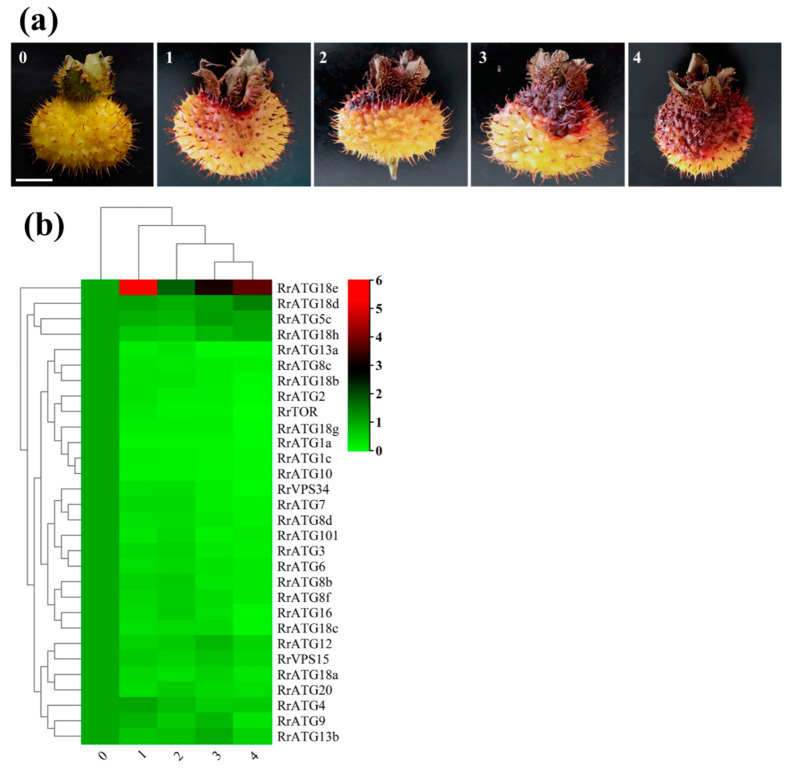
Schematic diagram and expression of *RrATGs* in fruits with different grades of TRD occurrence. (**a**) Grade 0 is no incidence, grade 1 is 1–10%, grade 2 is 11–25%, grade 3 is 26–50%, and grade 4 is >50% the proportion of fruit spot size to the total surface area of the fruit, respectively. Bar = 1 cm. (**b**) Heat map showed the corresponding expression levels of *RrATGs* in diseased fruits of different grades, and the expression levels of healthy fruits were considered as ‘1’, red and green represent the higher and lower expression levels, respectively.

**Figure 6 biomolecules-13-00556-f006:**
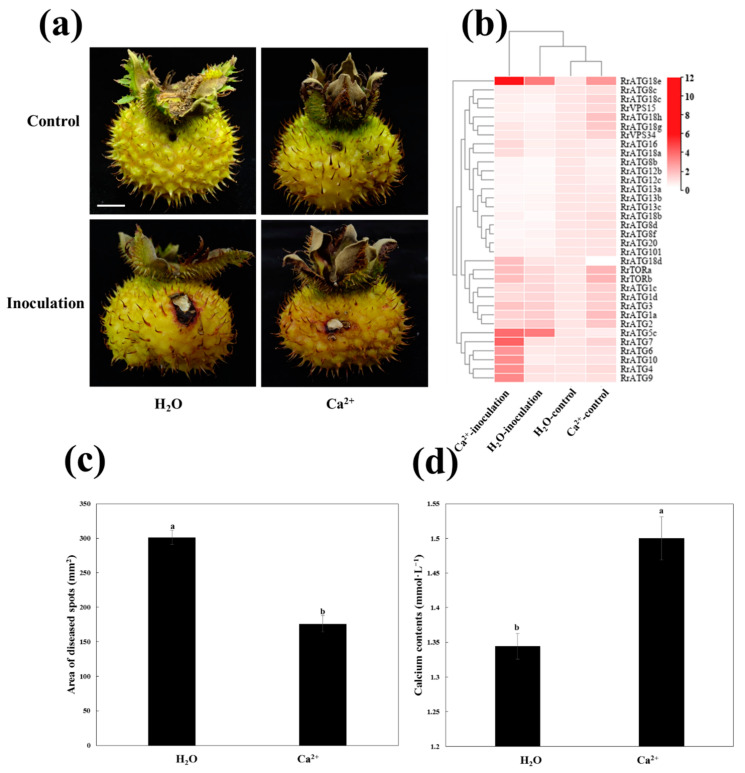
Schematic diagram of Ca^2+^ enhanced the resistance of *R. roxburghii* fruits to TRD. (**a**) Phenotypes of fruits inoculated with *C. fructicola* and control PDA after H_2_O and Ca^2+^ treatments at 13 DPI. Bar = 1 cm. (**b**) The heat map showed the expression of *RrATGs* under different treatments at 13 DPI. The H_2_O-control represented the expression of inoculated control PDA after H_2_O-treated, the H_2_O-inoculation represented the expression of inoculated *C. fructicola* PDA after H_2_O-treated, the Ca^2+^-control represented the expression of inoculated control PDA after Ca^2+^-treated, the Ca^2+^-inoculation represented the expression of inoculated *C. fructicola* PDA after Ca^2+^-treated. The expression of H_2_O-control was considered “1”, red and white represented the higher and lower expression levels, respectively. (**c**) The area of diseased spots of fruits inoculated with *C. fructicola* PDA after H_2_O and Ca^2+^ treatments at 13 DPI. (**d**) Total calcium content of fruits after H_2_O and Ca^2+^ treatments at 13 DPI. Data are means of standard errors of three replicates. The letters on the column denote significant differences (*p* < 0.05, ANOVA) between H_2_O and Ca^2+^ treatments.

**Table 1 biomolecules-13-00556-t001:** Related information of autophagy-related genes (ATGs) in *R. roxburghii*.

Gene Name	*Arabidopsisa* ID	Gene	*R. roxburghii* ID	Identity to *Arabidopsisa*(%)	Protein (aa)	Protein Molecular Mass(KDa)	pI	Predicted Localization
*AtATG1a*	At3g61960	*RrATG1a*	Contig179.812	64.78	722	79.88	6.73	Nuclear
*AtATG1b*	At3g53930	*RrATG1b*	Contig110.67	61.36	138	15.41	9.88	Nuclear
*AtATG1c*	At2g37840	*RrATG1c*	Contig191.2	52.21	677	74.79	6.44	Nuclear
*AtATG1d*	At1g49180	*RrATG1d*	Contig289.274	49.16	649	71.65	6.51	Nuclear
*At* *ATG2*	At3g19190	*Rr* *ATG2*	Contig161.356	49.80	1983	217.54	5.47	Plasma membrane
*At* *ATG3*	At5g61500	*Rr* *ATG3*	Contig189.150	82.86	366	41.44	4.51	Cytoskeleton
*At* *ATG4a*	At2g44140	*Rr* *ATG4*	Contig161.437	55.85	427	46.98	4.98	Chloroplast
*At* *ATG4b*	At3g59950	NA	NA	NA	NA	NA	NA	NA
*At* *ATG5*	At5g17290	*Rr* *ATG5a*	Contig361.91	61.98	362	40.95	4.61	Cytoplasmic
		*Rr* *ATG5b*	Contig8.24	59.53	302	33.96	5.21	Cytoplasmic
		*Rr* *ATG5c*	Contig169.100	58.92	310	35.38	5.99	Nuclear
*At* *ATG6*	At3g61710	*Rr* *ATG6*	Contig289.113	67.13	469	52.98	5.58	Cytoplasmic
*At* *ATG7*	At5g45900	*Rr* *ATG7*	Contig363.72	63.96	581	63.17	6.42	Endoplasmic reticulum
*At* *ATG8a*	At4g21980	*RrATG8a*	Contig179.203	63.53	110	11.91	5.02	Mitochondrial
*At* *ATG8b*	At4g04620	*RrATG8b*	Contig18.48	76.07	119	13.65	5.00	Cytoplasmic
*At* *ATG8c*	At1g62040	*RrATG8c*	Contig360.160	91.45	119	13.72	9.29	Cytoplasmic
*At* *ATG8d*	At2g05630	*RrATG8d*	Contig10.163	82.46	121	13.84	9.32	Cytoplasmic
*At* *ATG8e*	At2g45170	NA	NA	NA	NA	NA	NA	NA
*At* *ATG8f*	At4g16520	*RrATG8f*	Contig136.201	87.18	117	13.42	9.77	Cytoplasmic
*At* *ATG8g*	At3g60640	NA	NA	NA	NA	NA	NA	NA
*At* *ATG8h*	At3g06420	NA	NA	NA	NA	NA	NA	NA
*At* *ATG8i*	At3g15580	NA	NA	NA	NA	NA	NA	NA
*At* *ATG9*	At2g31260	*Rr* *ATG9*	Contig385.359	70.63	808	93.13	7.60	Plasma membrane
*At* *ATG10*	At3g07525	*Rr* *ATG10*	Contig290.80	68.85	525	58.91	8.42	Chloroplast
*At* *ATG11*	At4g30790	NA	NA	NA	NA	NA	NA	NA
*At* *ATG12a*	At1g54210	*Rr* *ATG12a*	Contig428.656	81.25	110	12.70	9.12	Nuclear
*At* *ATG12b*	At3g13970	*Rr* *ATG12b*	Contig414.84	80.85	95	10.76	10.11	Chloroplast
		*Rr* *ATG12c*	Contig401.201	80	77	8.73	10.06	Mitochondrial
		*Rr* *ATG12d*	Contig385.680	69.66	144	16.30	10.40	Chloroplast
*At* *ATG13a*	At3g49590	*RrATG13a*	Contig386.98	52.04	1032	115.66	9.07	Nuclear
*At* *ATG13b*	At3g18770	*RrATG13b*	Contig52.3	56.70	644	71.33	7.97	Cytoplasmic
		*RrATG13c*	Contig266.16	56.24	628	69.41	7.84	Cytoplasmic
*At* *ATG16*	At5g50230	*Rr* *ATG16*	Contig104.384	63.32	745	82.24	8.56	Nuclear
*At* *ATG18a*	At3g62770	*RrATG18a*	Contig385.717	89.50	928	104.33	6.46	Nuclear
*At* *ATG18b*	At4g30510	*RrATG18b*	Contig405.14	74.18	371	40.31	6.61	Extracellular
*At* *ATG18c*	At2g40810	*RrATG18c*	Contig317.6	75.37	411	45.67	7.54	Nuclear
*At* *ATG18d*	At3g56440	*RrATG18d*	Contig121.44	69.37	280	31.63	7.99	Golgi
*At* *ATG18e*	At5g05150	*RrATG18e*	Contig121.43	64	783	87.27	7.88	Cytoplasmic
*At* *ATG18f*	At5g54730	NA	NA	NA	NA	NA	NA	NA
*At* *ATG18g*	At1g03380	*RrATG18g*	Contig385.614	61.30	1243	136.39	7.08	Chloroplast
*At* *ATG18h*	At1g54710	*RrATG18h*	Contig149.53	63.74	863	94.06	5.637	Mitochondrial
*At* *ATG20*	At5g06140	*Rr* *ATG20*	Contig179.205	76.03	337	38.74	9.31	Chloroplast
*At* *ATG101*	At5g66930	*Rr* *ATG101*	Contig124.67	76.44	208	24.09	6.60	Cytoplasmic
*At* *TOR*	At1g50030	*Rr* *TORa*	Contig59.5	81.53	2459	276.22	6.83	Cytoplasmic
		*Rr* *TORb*	Contig161.4	79.85	2449	274.88	6.82	Cytoplasmic
*At* *VPS15*	At4g29380	*Rr* *VPS15*	Contig354.8	70.62	1555	174.30	7.27	Nuclear
*At* *VPS34*	At1g60490	*Rr* *VPS34*	Contig332.44	84.17	805	91.87	7.06	Cytoplasmic

## Data Availability

The datasets used or analyzed during the current study available from the corresponding author upon reasonable request.

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
