# Peer review of "Genome-Wide Identification and Expression Analysis of Rosa roxburghii Autophagy-Related Genes in Response to Top-Rot Disease"

_biomolecules, 2023, doi:10.3390/biom13030556_

Round 1

Reviewer 1 Report

The manuscript by Luo and collaborators identified and characterized 40 Rosa roxburghii autophagy-related genes (RsATGs). The authors detected the transcripts of these genes at different stages of development of R. roxburghii and during the occurrence of top-rot disease (TRD) caused by the fungus Colletotrichum fructicola. Furthermore, they demonstrated that exogenous calcium (Ca2+) increases RrATG18e mRNA levels and fruit resistance to TRD. The work presents important data to understand the relationship of RrATGs genes expression of R. roxburghii in the development and response of the plant to resistance to TRD caused by fungal infection. Overall the manuscript is well written and the figures are clear and informative. However, some questions should be considered:

Major concerns

1) The main problem of the article is its weak correlation between the expression of ATG genes (in particular RrATG18e) and their role in resistance to fungal infection (e.g. - page 13, lines 5-6). Although there is a relationship in the literature between the plant's immune response and autophagy, in general, the data in the article do not support this correlation, since there was no modulation of the RrATG genes (with the exception of RrATG18e). This needs to be clarified and changes to the text must be included in order to avoid misunderstanding.

2) Another problem is related to methodology. The RT-qPCR assay has only one normalizer. Although, generally UBI is a good normalizing gene, it is necessary to include at least one other housekeeping gene in these analyzes to confirm the observed profile. It is not clear from the manuscript whether the authors followed the Miqe (Minimum Information for Publication of Quantitative Real-Time PCR Experiments) guidelines for evaluating primer efficiency, and other parameters of qPCR reactions.

Minor revision

1) Page 3, lines 41-42 – the use of the term sterilize the fruits with 75% ethanol is not correct. Perhaps the authors should change to other terms, such as disinfection or decontamination.

2) Table 1 – the first row of the table is unformatted.

3) Table 1 – change the term Kda to KDa;

4) Table 1 - change the term Pi to pI.

5) Table 1 - It would be interesting for the authors to identify the subfamilies in the table, perhaps with different colors or with gray scale.

6) Page 8, line 8 – switch "Fifty four" to lowercase.

Author Response

Dear Reviewer 1,

Many thanks for your comments concerning our manuscript entitled “Genome-Wide Identification and Expression Analysis of Rosa roxburghii Autophagy-Related Genes in Response to Top-Rot Disease” (ID: biomolecules-2140818). We sincerely thank you very much for giving us an opportunity to revise our manuscript! We earnestly appreciate for your warm work! Your comments are all valuable and very helpful for revising and improving our manuscript, as well as the important guiding significance to our researches. We have studied carefully your comments and have made corresponding corrections which we hope meet with approval. Attached please find the revised manuscript. The followings are our point-by-point responses to the comments and suggestions raised by you.

Response to Reviewer 1 Comments

Major concerns

Point 1: The main problem of the article is its weak correlation between the expression of ATG genes (in particular RrATG18e) and their role in resistance to fungal infection (e.g. - page 13, lines 5-6). Although there is a relationship in the literature between the plant's immune response and autophagy, in general, the data in the article do not support this correlation, since there was no modulation of the RrATG genes (with the exception of RrATG18e). This needs to be clarified and changes to the text must be included in order to avoid misunderstanding.

Response 1:

Thank you very much for your careful review and your important comments! Our findings suggest that for fruits with different degrees of TRD occurrence, the expression of RrATG18e gene was significantly up-regulated in diseased fruits compared to healthy fruits (page 10, lines 1-5). The expression of RrATG18e gene was significantly up-regulated in fruits inoculated with C. fructicola PDA after H2O-treated compared to fruits inoculated with control PDA after H2O-treated (page 13, lines 8-11). The above-mentioned results show that the infection of C. fructicola induced up-regulation of RrATG18e gene expression in diseased fruits compared to healthy fruits. The specific role of RrATG18e in the resistance of R. roxburghii fruits to fungal infection remains to be verified in the next experiments. So we have took your suggestion, and revised the sentence on page 13, lines 16-18 in the revised manuscript as follows: “Therefore, RrATG18e might play a comparatively important role in calcium-mediated enhancement of the resistance of R. roxburghii fruits to TRD.“

Point 2: Another problem is related to methodology. The RT-qPCR assay has only one normalizer. Although, generally UBI is a good normalizing gene, it is necessary to include at least one other housekeeping gene in these analyzes to confirm the observed profile. It is not clear from the manuscript whether the authors followed the Miqe (Minimum Information for Publication of Quantitative Real-Time PCR Experiments) guidelines for evaluating primer efficiency, and other parameters of qPCR reactions.

Response 2:

Special thanks to you for your careful review and your valuable suggestion! Firstly, we have been researching the species of R. roxburghii for many years, and we have screened many housekeeping genes, among which UBQ is relatively the most stable. The other conventional housekeeping genes are not very stable, so we only chose this one. Secondly, we performed the qPCR reactions strictly according to the Miqe guidelines, as described in the red-labeled section on page 4, lines 11-21 in the revised manuscript for additional details.

Minor revision

1) Page 3, lines 41-42 – the use of the term sterilize the fruits with 75% ethanol is not correct. Perhaps the authors should change to other terms, such as disinfection or decontamination.

Response 1):

Thank you very much for your careful review and your suggestion! The sentence was modified in the revised manuscript: Firstly, the surface was disinfected with 75% ethanol for 15 minutes and then washed with sterile water. (See page 3, lines 41-42)

2) Table 1 – the first row of the table is unformatted.

Response 2):

Thank you a lot for your valuable suggestion! The first row of the table has been formatted. (See pages 5-6)

3) Table 1 – change the term Kda to KDa;

Response 3):

Many thanks to you for your hard work and your professional suggestion! “Kda” has been modified to “KDa”, which was marked in red in the revised manuscript. (See pages 5-6)

4) Table 1 - change the term Pi to pI.

Response 4):

Thank you very much for your careful review and your suggestion! “Pi” has been modified to “pI”, which was marked in red in the revised manuscript. (See pages 5-6)

5) Table 1 - It would be interesting for the authors to identify the subfamilies in the table, perhaps with different colors or with gray scale.

Response 5):

Thank you very much for your attention to our manuscript! We will take your suggestion under advisement. We hope to have the opportunity to discuss it with you again. But, it may be slightly confusing if different colors are used to identify the subfamilies in the Table 1, as there are 19 subfamilies identified. Thank you most sincerely!

6) Page 8, line 8 – switch "Fifty four" to lowercase.

Response 6):

Many thanks to you for your careful review and your suggestion! “Fifty four” has been modified to “fifty four”, which was marked in red in the revised manuscript. (See page 7, line 24)

As mentioned above, we tried our best to improve the manuscript and made some changes in the revised manuscript. These changes will not influence the content and framework of the manuscript. And here we did not list all the changes but mark them in red in the body of the revised manuscript.

We appreciate for reviewers and editors' warm work earnestly, and hope that the correction will meet with approval.

Once again, thank you very much and best regards!

Reviewer 2 Report

In this study, 

Kaisha Luo and authors reported genome-wide identification and expression analysis of Rosa roxburghii autophagy-related genes when infected with a causal agent of top-rot. The study is very informative. The data analysis is in accordance with the conclusions. The writing was easy to follow. My only suggestion for authors is to present  more clearly the results of analysis of the plant samples obtained from orchards of Chaxiang Village, Gujiao Town, Longli

County, Guizhou Province, China, and those obtained from pathogen inoculation trials  conducted in the fruit germplasm repository of Guizhou University, Guizhou, China.

The manuscript would also benefit if authors add in the Discussion section opinion on how other abiotic environmental factors could affect main findings. In general, I would recommend the acceptance of the manuscript.  

Author Response

Dear Reviewer 2,

Many thanks for your comments concerning our manuscript entitled “Genome-Wide Identification and Expression Analysis of Rosa roxburghii Autophagy-Related Genes in Response to Top-Rot Disease” (ID: biomolecules-2140818). We sincerely thank you very much for giving us an opportunity to revise our manuscript! We earnestly appreciate for your warm work! Your comments are all valuable and very helpful for revising and improving our manuscript, as well as the important guiding significance to our researches. We have studied carefully your comments and have made corresponding corrections which we hope meet with approval. Attached please find the revised manuscript. The followings are our point-by-point responses to the comments and suggestions raised by you.

Response to Reviewer 2 Comments

In this study, Kaisha Luo and authors reported genome-wide identification and expression analysis of Rosa roxburghii autophagy-related genes when infected with a causal agent of top-rot. The study is very informative. The data analysis is in accordance with the conclusions. The writing was easy to follow. My only suggestion for authors is to present more clearly the results of analysis of the plant samples obtained from orchards of Chaxiang Village, Gujiao Town, Longli County, Guizhou Province, China, and those obtained from pathogen inoculation trials conducted in the fruit germplasm repository of Guizhou University, Guizhou, China.

The manuscript would also benefit if authors add in the Discussion section opinion on how other abiotic environmental factors could affect main findings. In general, I would recommend the acceptance of the manuscript.

Response:

Special thanks to you for your careful reviews and the positive comments on our manuscript! Your comments are all valuable and have been very helpful for revising and improving our manuscript. Based on your comments, we have made corresponding changes to our manuscript to further clarifying the results of analysis of the plant samples obtained from orchards of Chaxiang Village, Gujiao Town, Longli County, Guizhou Province, China, and those obtained from pathogen inoculation trials conducted in the fruit germplasm repository of Guizhou University, Guizhou, China, which were marked in red in the revised manuscript. (See pages 9-10, 13)

According to your suggestion, have added opinions into the Discussion section on how other abiotic environmental factors could affect main findings in the revised manuscript, which were marked in red. (See page 14)

As mentioned above, we tried our best to improve the manuscript and made some changes in the revised manuscript. These changes will not influence the content and framework of the manuscript. And here we did not list all the changes but mark them in red in the body of the revised manuscript.

We appreciate for reviewers and editors' warm work earnestly, and hope that the correction will meet with approval.

Once again, thank you very much and best regards!

Reviewer 3 Report

Manuscript "Genome-Wide Identification and Expression Analysis of Rosa roxburghii Autophagy-Related Genes in Response to Top-Rot Disease" is very interesting.

General comments:
Authors provided new insight into the potential role of RrATGs in the defensive responses of R. roxburghii fruits to TRD.
Bioinformatics analyses are perfect.
Statistical analyses are correct.

Detailed comments:
Figure 6b: There is no information about the grouping method used and the construction of the figure.
Figures 6c and 6d: need HSD values.

Paper needs minor revision.

Author Response

Dear Reviewer 3,

Many thanks for your comments concerning our manuscript entitled “Genome-Wide Identification and Expression Analysis of Rosa roxburghii Autophagy-Related Genes in Response to Top-Rot Disease” (ID: biomolecules-2140818). We sincerely thank you very much for giving us an opportunity to revise our manuscript! We earnestly appreciate for your warm work! Your comments are all valuable and very helpful for revising and improving our manuscript, as well as the important guiding significance to our researches. We have studied carefully your comments and have made corresponding corrections which we hope meet with approval. Attached please find the revised manuscript. The followings are our point-by-point responses to the comments and suggestions raised by you.

Response to Reviewer 3 Comments

Detailed comments:

1) Figure 6b: There is no information about the grouping method used and the construction of the figure.

Response 1):

Many thanks to you for your careful review and hard work! On the basis of your suggestion, the information about the grouping method used and the construction of the figure have been added, which were marked in red in the revised manuscript. (See page 12)

2) Figures 6c and 6d: need HSD values.

Response 2):

Thank you a lot for your hard work and your careful review! Based on your suggestion, HSD values have been added in Figures 6c and 6d. (See page 12)

3) Paper needs minor revision.

Response 3):

Thank you very much for your attention to our manuscript! We have studied carefully your comments and have made corrections of the format and grammar which were marked in red in the revised manuscript.

As mentioned above, we tried our best to improve the manuscript and made some changes in the revised manuscript. These changes will not influence the content and framework of the manuscript. And here we did not list all the changes but mark them in red in the body of the revised manuscript.

We appreciate for reviewers and editors' warm work earnestly, and hope that the correction will meet with approval.

Once again, thank you very much and best regards!

Round 2

Reviewer 1 Report

All questions were properly addressed.